# Properties of Fiber Bragg Grating in CYTOP Fiber Response to Temperature, Humidity, and Strain Using Factorial Design

**DOI:** 10.3390/s22051934

**Published:** 2022-03-01

**Authors:** Ying-Gang Nan, Nazila Safari Yazd, Ivan Chapalo, Karima Chah, Xuehao Hu, Patrice Mégret

**Affiliations:** 1The Electromagnetism and Telecommunication Department, University of Mons, 31 Boulevard Dolez, 7000 Mons, Belgium; nazila.safariyazd@umons.ac.be (N.S.Y.); ivan.chapalo@umons.ac.be (I.C.); karima.chah@umons.ac.be (K.C.); patrice.megret@umons.ac.be (P.M.); 2Research Center for Advanced Optics and Photoelectronics, Department of Physics, College of Science, Shantou University, Shantou 515063, China; xhhu3@stu.edu.cn

**Keywords:** fiber bragg gratings, polymer optical fiber, CYTOP, factorial design

## Abstract

The characteristics of fiber Bragg grating (FBG) in cyclic transparent fluoropolymer (CYTOP) optical fiber have attracted more and more attention in recent years. However, different results of the FBG response to environmental parameters are reported. This work presents a three-variable two-level factorial experimental method to investigate the FBG response to temperature, humidity, and strain in CYTOP fiber. Two uniform FBGs are inscribed separately in CYTOP fiber with and without over-clad. With only eight measuring points, the interactions among three variable parameters are computed and the parameter sensitivities and cross-sensitivities are estimated. Similar temperature and strain sensitivities were found for both gratings, whereas significant cross-sensitivity between humidity and temperature was present only in FBG inscribed in CYTOP fiber with over-clad.

## 1. Introduction

Among the continued development of sensing techniques, fiber Bragg grating (FBG) is one of the most efficient and convenient technologies due to its unique characteristics such as light weight, small size, and high sensitivity [1]. The FBG is a periodic modulation of the refractive index in the fiber core, which leads to the reflection by the grating planes of a specific wavelength, referred to as the Bragg wavelength, whereas the remaining wavelengths pass through when light propagates into the FBG. The reflected wavelengths should satisfy the phase matching conditions, which depend on the effective refractive index and the grating period [1]. When the surrounding environmental variables are changed, the phase matching conditions are modified and this results in the change of the reflected wavelengths, i.e., the Bragg wavelength undergoes a wavelength shift that can be used to measure a sensing parameter. To date, this sensing principle has been used, for example, to monitor temperature [2], humidity [3], strain [4], vibration [5], surrounding refractive index [6], and gamma radiation [7].

Due to significant properties such as low Young’s modulus, biocompatibility, low weight, and high flexibility, polymer optical fibers (POFs) received more and more attention in the fields of sensing applications [8,9,10]. Among the POFs, CYTOP fiber is a good candidate due to its significantly low attenuation in telecom transparency windows and specific material characteristics [11,12,13,14,15]. For fiber Bragg gratings fabrication in CYTOP fibers, two inscription methods have been used to date. One is the direct inscription technique that generally uses a femtosecond pulsed laser at 517 nm [13,16] or 800 nm [12]. Although high reflectivity gratings are achieved by this method, the direct inscription process requires expensive, high-precision automated translation stages. Another way to fabricate FBGs in CYTOP is the phase mask technique [17,18] using the Krypton Fluoride (KrF) excimer laser at 248 nm. Gratings fabricated by this excimer laser show relatively low reflectivities, broad reflection peaks (9 nm to 10 nm) and require long exposure time (up to 60 min) [17,18]. It is worth to note that the over-clad of the CYTOP should be removed when using the excimer laser [17,18], whereas the over-clad is preserved when the direct inscription method is used [12,13,16]. In summary, there are two different CYTOP-FBG structures, one with over-clad, and one without the over-clad. In this paper, we use a setup combining the phase mask technique and a femtosecond pulsed laser at 400 nm to produce, with good repeatability, high reflectivity FBGs in CYTOP fibers with and without over-clad in a few seconds.

Papers on CYTOP fibers disclose some experimental results on CYTOP-FBG sensitivities to temperature, humidity and strain [8,11,12,17,18], but with significant dispersion in the computed sensitivity values, especially for the temperature sensitivity that ranges from 17.6 pm/°C to 37.7 pm/°C. Moreover, there is no detailed analysis of the cross-effects between temperature, humidity and strain. Therefore, the aim of this paper is to design a set-up able to experimentally investigate in depth the sensitivities and, more importantly, the cross-sensitivities of CYTOP-FBGs with and without over-clad in response to temperature, humidity, and strain. To achieve that goal in an efficient way, we use a three-variable, two-level factorial experiment, specially designed to characterize the CYTOP-FBG with three simultaneous stimuli (temperature, humidity and strain).

## 2. Theory of Factorial Design

The classical experimental approach to measure the effect of multi-stimuli in fiber Bragg grating sensors is to study each variable separately [18]. However, this approach cannot reveal the interactions between the factors (parameters). For example, with 3 factors and 10 levels per factor, 3×10=30 experiments need to be carried out which is time consuming and prone to error.

To solve this problem, a design of experiment (DoE) method based on a factorial design is a better strategy and much more time effective. The advantages of the DoE are: (1) the total number of experiments is reduced, and (2) the interactions between the factors can be easily computed [19,20].

In general, in the factorial method, the number of variables (factors) is *k*, and *n* levels are imposed per factor, leading to $n^{k}$ experiments. The number of experiments becomes quickly very high if *n* is large. To minimize the measuring time, the minimum choice for *n* is two, and for three variables this scheme is referred to as a two-level, three-variable design. This design will correctly operate provided that the eight measuring points span the entire experimental range of the three variables, and a mathematical model can be used to interpolate the results between the measuring points.

For three variables with two levels per variable, each factor $X_{i}$ has its range $\left[X_{i,\min},X_{i,\max}\right]$ and its unit. To ease computation and comparison, it is common to denote the low levels by −1 and the high levels by +1, and to use centered and scaled variables $x_{i}$ defined by the relation
(1)$$x_{i}=\frac{X_{i}-m_{i}}{\Delta_{i}}$$
where $m_{i}=\left(X_{i,\max}+X_{i,\min}\right)/2$ is the central value and $\Delta_{i}=\left(X_{i,\max}-X_{i,\min}\right)/2$ is the step. Then, the best positions of the experimental points are located at the vertices (minimal and maximal values of $X_{i}$, or $x_{i}=\pm 1$, named a, b, c, d, e, f, g and h in Figure 1) of the experimental cube describing the ranges of the parameters, and the mathematical model is the linear model with first-order interactions, i.e., the Bragg wavelength $\lambda_{B}$ varies according to the expression
(2)$$\lambda_{B}=a_{0}+a_{T}x_{T}+a_{H}x_{H}+a_{S}x_{S}+a_{TH}x_{T}x_{H}+a_{TS}x_{T}x_{S}+a_{HS}x_{H}x_{S}+a_{THS}x_{T}x_{H}x_{S}$$
where the coefficients *a* represent the normalized sensitivities and the cross-sensitivities.

In this study, the three factors are the temperature ($X_{T},x_{T}$), the humidity ($X_{H},x_{H}$), and the strain ($X_{S},x_{S}$). The maximal safe operating temperature of the CYTOP-FBGs is reported to be 60 °C [5]. To avoid damaging the CYTOP-FBGs, we limit the maximum temperature characterization to 45 °C. Besides, CYTOP-FBG without over-clad has a diameter of 50 μm and therefore shows weaker mechanical properties compared to over-clad fibers with an external diameter of 490 μm. We experimentally found that the CYTOP fiber without over-clad can sustain a strain up to 571 µε at room temperature. To avoid any unwanted or irreversible mechanical deformation, the fiber was manipulated with caution and we limited the strain range to 155 µε. In this work, the low and high level of temperature, relative humidity and strain are 25 °C to 45 °C, 30%RH to 90%RH, and 0 µε to 155 µε, respectively, as shown in Table 1.

Table 2 displays the parameters ($x_{T},x_{H},x_{S}$) of the 8 measurements $\lambda_{i},i=a,\ldots ,h$ to be performed, as well as the values of the interactions two by two ($x_{T}x_{H}$, $x_{T}x_{S}$, $x_{H}x_{S}$), and three by three ($x_{T}x_{H}x_{S}$). The column *I* is simply a column of 1.

Relation (Equation 2) contains 8 unknown coefficients *a* that can be computed from the 8 measurements $\lambda_{a}$ to $\lambda_{h}$ (see Figure 1). Indeed, if Equation (Equation 2) is written for each $\lambda_{i},i=a,\ldots ,h$, the following matrix relation holds [21]
(3)λ=Xa
where λ is the 8×1 measurement matrix, X is a 8×8 matrix called the model matrix (made from columns *I* to $x_{T}x_{H}x_{S}$ of Table 2) and a is the matrix of the unknown coefficients, respectively equal to:

$\lambda =\left[\begin{array}{c} \lambda_{a}\\ \lambda_{b}\\ \lambda_{d}\\ \lambda_{c}\\ \lambda_{e}\\ \lambda_{f}\\ \lambda_{h}\\ \lambda_{g} \end{array}\right]$, $X=\left[\begin{array}{cccccccc} +1 & -1 & -1 & -1 & +1 & +1 & +1 & -1\\ +1 & +1 & -1 & -1 & -1 & -1 & +1 & +1\\ +1 & -1 & +1 & -1 & -1 & +1 & -1 & +1\\ +1 & +1 & +1 & -1 & +1 & -1 & -1 & -1\\ +1 & -1 & -1 & +1 & +1 & -1 & -1 & +1\\ +1 & +1 & -1 & +1 & -1 & +1 & -1 & -1\\ +1 & -1 & +1 & +1 & -1 & -1 & +1 & -1\\ +1 & +1 & +1 & +1 & +1 & +1 & +1 & +1 \end{array}\right]$, and $a=\left[\begin{array}{c} a_{0}\\ a_{T}\\ a_{H}\\ a_{S}\\ a_{TH}\\ a_{TS}\\ a_{HS}\\ a_{THS} \end{array}\right]$.

The way the matrix X is constructed makes it automatically non-singular so that it can be inverted to give the coefficients *a* by the simple matrix product of Equation (Equation 4)
(4)$$a=X^{-1}\lambda =\frac{1}{8}X^{T}\lambda$$
where $X^{T}$ is the transpose matrix of X. It is therefore very simple to compute the 8 normalized coefficients *a* from the 8 measurements, and then use relation (Equation 2) to interpolate the Bragg wavelength for any point inside the experimental domain. Moreover, it is also straightforward to denormalize the coefficients to get:(5)$$\begin{array}{c} \lambda_{B}=A_{0}+A_{T}\Delta X_{T}+A_{H}\Delta X_{H}+A_{S}\Delta X_{S}+A_{TH}\Delta x_{T}\Delta x_{H}+\\ A_{TS}\Delta x_{T}\Delta X_{S}+A_{HS}\Delta X_{H}\Delta X_{S}+A_{THS}\Delta X_{T}\Delta X_{H}\Delta X_{S} \end{array}$$
where $\Delta X_{i}=X_{i}-m_{i}$ is the variation of the factor $X_{i}$ around its mean value $m_{i}$ and the links between *a* and *A* are expressed by the following relations
(6)$$A_{0}=a_{0},A_{i}=\frac{a_{i}}{\Delta_{i}},A_{ij}=\frac{a_{ij}}{\Delta_{i}\Delta_{j}}\;\mathrm{and}\;A_{ijk}=\frac{a_{ijk}}{\Delta_{i}\Delta_{j}\Delta_{k}}$$

Physically, the coefficient $A_{i}$ is the sensor sensitivity around the center (point *Q* in Figure 1) for the factor $X_{i}$, whereas $A_{ij}$ is the sensor cross-sensitivity around point *Q* for the interaction of factors $X_{i}$ and $X_{j}$. Finally, $A_{0}$ corresponds to the Bragg wavelength at the cube center, point *Q*.

## 3. Fiber Bragg Grating Inscription

In this work, a femtosecond pulsed laser (a Mai Tai and Spitfire Pro. amplifier from Spectra Physics) producing 120 fs light pulses at 800 nm with a repetition rate of 1 kHz and energy of 4 mJ is used as the inscription laser. Gratings in CYTOP fibers are inscribed with the second harmonic generated pulses at 400 nm built in a nonlinear crystal starting from the 800 nm original pulses. The pulses then pass through a phase mask with a period of 1158 nm before reaching the CYTOP fiber with and without over-clad. The combination of the fs pulsed laser and the phase mask technique provides an efficient inscription method, producing FBGs with high reflectivity and good repeatability. It is important to mention that for the FBG in CYTOP without over-clad, the polycarbonate-based over-clad is removed on a length of 2 cm by dichloromethane CH_2_Cl_2_ in a few minutes [11,22,23]. On the other hand, for the CYTOP fiber with the over-clad, a cover glass (from Corning) is inserted between the phase mask and the CYTOP fiber to protect the phase mask [24]. The commercially available graded-index CYTOP fibers (GigaPOF-50SR and GigaPOF-62SR from Chromis) are used in this fabrication, with core diameter, over-clad diameter, and numerical aperture of 50 μm, 490 μm, and 0.185, and 62.5 μm, 490 μm, and 0.185, respectively [25,26]. We inscribe FBG through the over-clad in the 62.5 μm core diameter fiber, and we remove the over-clad for the for 50 μm core diameter fiber. Indeed, for the 50 μm core diameter fiber, it is difficult to get good quality FBGs by writing through the over-clad. As the thickness of the over-clad is larger, the optical intensity needed to inscribe FBGs is too high and creates damage to the structure.

The blue solid lines in Figure 2 represents the experimental FBG spectra of grating #1 and grating #2 in CYTOP fiber with and without over-clad, respectively.

These results are compared with the theory of gratings in multimode fibers (dotted lines). It is well-known that modes in parabolic graded index fibers propagate in M=V/2 groups, referred to as mode groups [11,27,28], each mode group being characterized by its effective refractive index $n_{m}$ given by the formula
(7)$$n_{m}=n_{\mathrm{co}}\sqrt{1-4m\Delta /V}.$$
where $n_{\mathrm{co}}$ is the refractive index of the core center, $\Delta =\left(n_{\mathrm{co}}^{2}-n_{\mathrm{cl}}^{2}\right)/2n_{\mathrm{co}}^{2}$ is the relative refractive index difference between core and cladding, and the V=2πaNA/λ is the normalized frequency with $\mathrm{NA}=\sqrt{n_{\mathrm{co}}^{2}-n_{\mathrm{cl}}^{2}}=n_{\mathrm{co}}\sqrt{2\Delta }$ the numerical aperture and *a* the core radius.

Then, from the fiber Bragg grating theory [1], the resonance wavelength $\lambda_{m}$ of the *m*th mode group of a grating with period Λ is computed by the grating equation
(8)$$\lambda_{m}=2n_{m}\Lambda$$

For 62.5 μm and 50 μm core diameter CYTOP fibers used in this study, the fiber specifications give NA≈0.185, $n_{\mathrm{co}}\approx 1.342$, a=31.25 μm and 25 μm. The period of grating is Λ=579 nm by fabrication, leading to V≃23.4 and 18.7, and M≃11 and 9 for λ around 1550 nm, respectively. The theoretically estimated values of the effective index and the resonance wavelength of each mode group are summarized in Table 3 and displayed as dotted lines in Figure 2. It clearly appears that the correspondence between the experimental resonance wavelengths and the theoretical ones is very good.

Another way to check theory and experiment is the agreement between the theoretical wavelength spacing of two consecutive mode groups given by the relation [11,27]
(9)$$\Delta \lambda =\frac{\lambda^{2}\mathrm{NA}}{2\pi an_{\mathrm{co}}^{2}}.$$

Applying relation (Equation 9) to our fibers gives 1.24 nm for grating #1 and 1.57 nm for grating #2, and the corresponding experimental values of 1.23 nm and 1.54 nm for gratings #1 and #2, respectively.

Figure 2 clearly shows the multi-peak like FBG spectra, typical of grating in multimode fibers, and the main question is “which mode group should be followed to get the measurement?”. Indeed, there are more than one peak that can be used for sensing. Nevertheless, it has been shown experimentally that the temperature sensitivities of the FBG are independent on the monitored peaks in graded-index multimode CYTOP fiber [11]. Therefore, in this work, we monitor the first mode group peak (labeled as $\lambda_{1}$) to analyze the properties of the CYTOP-FBGs.

## 4. Experimental Set-Up

Figure 3 represents the experimental set-up to measure the FBG response to the temperature, humidity and strain in CYTOP-FBGs.

A climate chamber (Weiss SB 22 [29]) is used to accurately set and control the temperature and humidity during the experiment. The climate chamber temperature and relative humidity ranges are −40 °C to 180 °C and 10%RH to 98%RH, respectively. Different calibrated masses were fixed, through a pulley system (see Figure 3), to the free end of the fiber to create inside the fiber a controlled strain given by the relation
(10)$$S=\frac{mg}{E_{\mathrm{CYTOP}}C_{\mathrm{CYTOP}}+E_{\mathrm{poly.}}C_{\mathrm{poly.}}}$$
where *m* is the mass of the load, *g* is the gravity acceleration (9.81 m/s^2^), $E_{\mathrm{CYTOP}}$ (1.3 GPa [30]) and $E_{\mathrm{poly.}}$ (2.2 GPa [31]) are the Young’s modulus of CYTOP and polycarbonate respectively, and $C_{\mathrm{CYTOP}}$ and $C_{\mathrm{poly.}}$ are the areas of the cross-sections of the fiber core and the over-clad, respectively.

The strain set-up is installed inside the climate chamber, whereas the interrogator (FS2200 from FiberSensing with a spectral resolution of 1 pm and a spectral range from 1500 nm to 1600 nm) and the control computer are placed outside of the climate chamber. A 3D stage is used to connect the CYTOP fiber and fiber pigtail (SMF-28). Moreover, a small drop of refractive index matching gel (n=1.4646 at 589.3 nm) is used between the two optical fibers to reduce the Fresnel reflections from the interfaces of the SMF and CYTOP [11].

The values of temperature, humidity, and strain corresponding to the measuring points in Figure 1 are set according to Table 1 and Table 2 to investigate the FBG characteristics in CYTOP with and without over-clad. The FBG spectra are recorded 4 h after setting up each humidity level and 30 min after each temperature value. More stabilization time is used primarily in the change of the humidity levels to make sure the humidity inside the CYTOP fiber is in equilibrium with the environmental humidity.

## 5. Results

The measured Bragg wavelengths of the two gratings corresponding to the eight measuring points are summarized in the Table 4.

Then using Equations (Equation 4)–(Equation 6), the normalized coefficients *a* and the denormalized coefficients *A* are computed and presented in Table 5 and Table 6, respectively.

Figure 4 shows the bar chart for the normalized main sensitivities and cross-sensitivities for the grating with over-clad (blue bar) and without over-clad (orange bar).

From this chart, many interesting observations are made: the most important factor for the FBG with over-clad is the humidity and then it is the temperature, whereas the FBG without over-clad is humidity insensitive. The temperature sensitivities with or without over-clad are similar, but there is strong interaction temperature-humidity only for the FBG with over-clad. The strain sensitivities are nearly identical, and there is no interaction with the other factors.

For the fibers used in this study, the over-clad material is polycarbonate, that is known to exhibit hydrophilicity, whereas CYTOP material shows hydrophobicity [32,33]. It is the reason why FBG with over-clad exhibits a high humidity sensitivity, but also a strong interaction between temperature and humidity. It is therefore vital, for temperature measurement, to compensate the humidity effect by a careful calibration of the sensor.

It is also worth noting that the temperature-humidity interaction of FBG-CYTOP with over-clad shows similar behavior to that of polyimide coated FBGs in silica fiber [19,20].

All the conclusions made up to now rely on the hypothesis that the linear model with first-order interaction of relation (Equation 2) is sufficient to describe the grating behavior for the three factors temperature, humidity and strain. To verify this assumption, some control points are added to the experiment, as shown in Figure 1. Seven points are used and chosen at the center of the cube (Q) and at the centers of cube faces ($S_{1}$ to $S_{6}$). The values of the Bragg wavelengths obtained at these control points are presented in Table 7, as well as the values computed from the model described by Equation (Equation 5) with the coefficients *A* of Table 6.

Figure 5 and Figure 6 translate in graphs the results of Table 7. In these figures, the blue points represents the wavelength differences between the control point measurements and the corresponding values obtained from the model described by Equation (Equation 5) (marked as red dots). The computation results are in good agreement with the experimental ones as they are within the error bar (blue line) of the experiment obtained from a series of repeated measurements and estimated to ±5 pm. For the grating with over-clad (Figure 5), the Bragg wavelength differences are below the resolution limit of the FBG interrogator (1 pm) for the measuring points $S_{1}$, $S_{5}$, $S_{6}$ and Q, while the difference is less than 5 pm for the $S_{2}$, $S_{3}$ and $S_{4}$. For the grating without over-clad the difference is always less than 5 pm. In both cases, the differences are within the error margins.

## 6. Discussion

Compared to the classical experimental method, the use of DoE to investigate the properties of CYTOP-FBGs is a powerful tool that gives a considerable amount of information with a minimum number of experiments. In this particular case, two levels per factor were sufficient to assess the behavior of the grating under temperature, humidity and strain stimuli.

From relation (Equation 5) with the coefficients *A* of Table 6, it is now possible to fully characterize the grating response inside the experimental domain depicted by the cube in Figure 1. For example, Figure 7a shows the temperature response of FBG #1 (with over-clad) for a strain of 50 µε and for different humidity levels, and Figure 7b displays the humidity response of FBG #1 for the same strain and different temperature levels. The cross-sensitivity temperature-humidity is clearly visible in these graphs, with a temperature sensitivity ranging from 27.7 pm/°C to 33.5 pm/°C for humidity levels of 30%RH and 90%RH, respectively.

Similarly, Figure 8a shows the temperature response of FBG #2 (without over-clad) for a strain of 50 µε and for different humidity levels, and Figure 8b displays the humidity response of FBG #1 for the same strain and different temperature levels. The cross-sensitivity temperature-humidity is clearly absent in these graphs, with a temperature sensitivity of 27.7 pm/°C for all humidity levels.

To date, different temperature sensitivities are reported in the literature [11,12,18,34]. These articles can be classified as CYTOP-FBGs with and without over-clad, respectively. For CYTOP-FBGs with over-clad [12,34], the temperature sensitivities are 17.62 pm/°C [34] and 37.7 ± 3.5 pm/°C [12] under relative humidity of 23%RH and 90%RH, respectively. The temperature sensitivity is lower for low humidity levels, and vice versa. These results are qualitatively in good agreement with the humidity dependent temperature sensitivity property of CYTOP-FBGs with the over-clad, as explained by Equation (Equation 5).

For CYTOP-FBGs without over-clad, temperature sensitivities of 27.5 pm/°C [11] and 27.5 ± 24 pm/°C [18] are reported for humidity levels of 80%RH and 50%RH, respectively. These results are very similar to the ones of this paper for FBG #2, showing a temperature sensitivity of 27.7 pm/°C independent of the humidity level.

The CYTOP-FBG sensitivity responses to strain are independent of humidity and temperature as the normalized interaction coefficients are around $10^{-3}$ for both gratings. Therefore there is nearly no difference in sensitivity with over-clad (1.46 pm/µε) and without over-clad (1.60 pm/µε), the difference being probably due to the different core diameters of the fibers. These values of strain sensitivities are again comparable to those found in the literature, 1.44 pm/µε [34] with over-clad, and 1.50 pm/µε [18] without over-clad.

## 7. Conclusions

The sensitivities and the cross-sensitivities of temperature, humidity, and strain were investigated in CYTOP-FBGs with and without the over-clad using a three-variable two-level factorial experimental method.

Gratings, in CYTOP fibers with and without over-clad, were fabricated using the phase mask technique and a femtosecond pulsed laser operating at 400 nm. Then gratings were spectrally characterized to check the multi-peak-like spectra typical of gratings in multimode fibers. The results of such measurements were found in good agreement with the theory.

After gratings fabrication, they were submitted to temperature, humidity and strain variations in a controlled environment to experimentally measure the main sensitivities and cross-sensitivities. From only eight measurements, the sensing properties of CYTOP-FBGs were fully determined, i.e., the eight coefficients of the linear model with first-order interactions were computed, and this allowed to estimate the sensing response for any point in the experimental domain. Of course, the validity of the linear model with first-order interaction was checked by adding seven control points carefully located in the experimental domain.

Finally, from this validated model, temperature, humidity and strain sensitivities of CYTOP-FBGs with and without over-clad were computed, demonstrating that the temperature sensitivity of FBG with over-clad is strongly affected by humidity, whereas it is humidity independent for FBG without over-clad. This is due to the hydrophilicity of the polycarbonate material used for the over-clad. On the other hand, strain sensitivity is nearly not affected by the humidity and temperatures levels for both gratings.

## Figures and Tables

**Figure 1 sensors-22-01934-f001:**
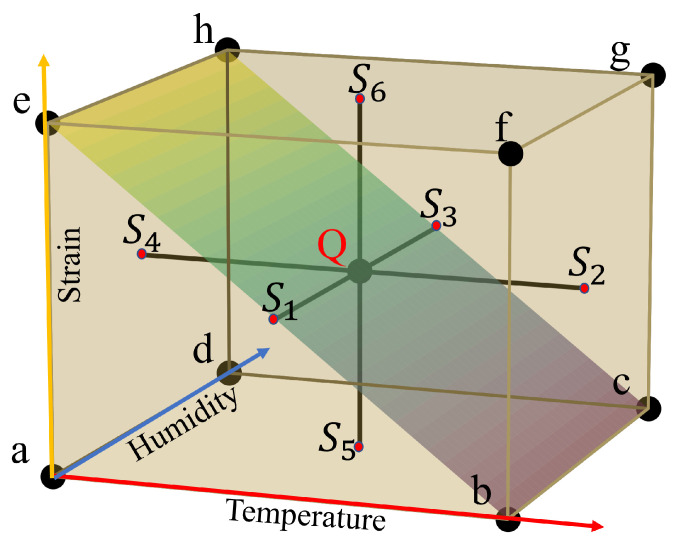
The experimental cube domain in a three-factor two-level factorial design.

**Figure 2 sensors-22-01934-f002:**
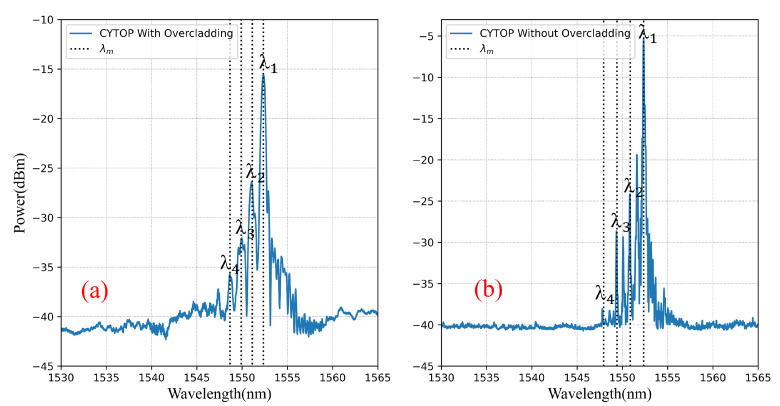
Reflection spectra for two gratings: (**a**) grating #1 in 62.5 μm core diameter CYTOP with over-clad (inscription parameters: grating length 8 mm, laser power 500 μW, and inscription time 20 s); (**b**) grating #2 in 50 μm core diameter CYTOP without over-clad (inscription parameters: grating length 5 mm, laser power 100 μW, and inscription time 10 s).

**Figure 3 sensors-22-01934-f003:**
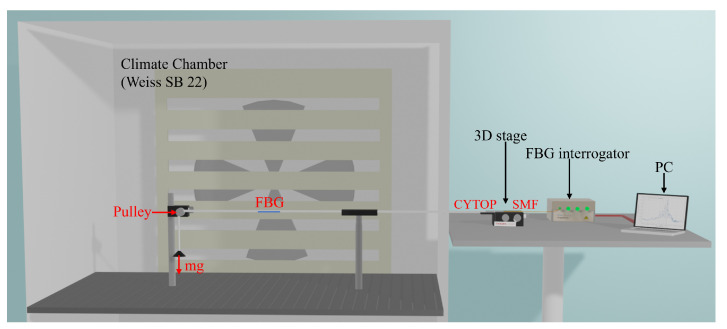
The experimental setup of CYTOP-FBG response to temperature, humidity and strain.

**Figure 4 sensors-22-01934-f004:**
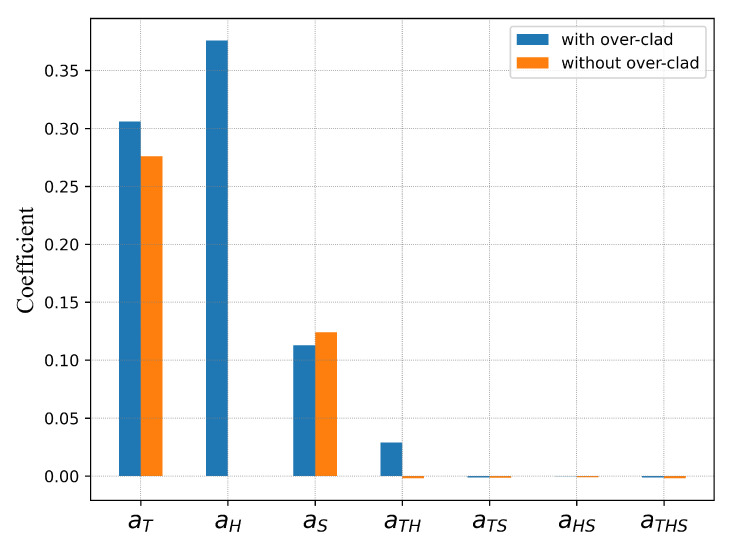
Relative distribution of the normalized main and cross-interaction sensitivities in CYTOP fiber with and without the over-clad.

**Figure 5 sensors-22-01934-f005:**
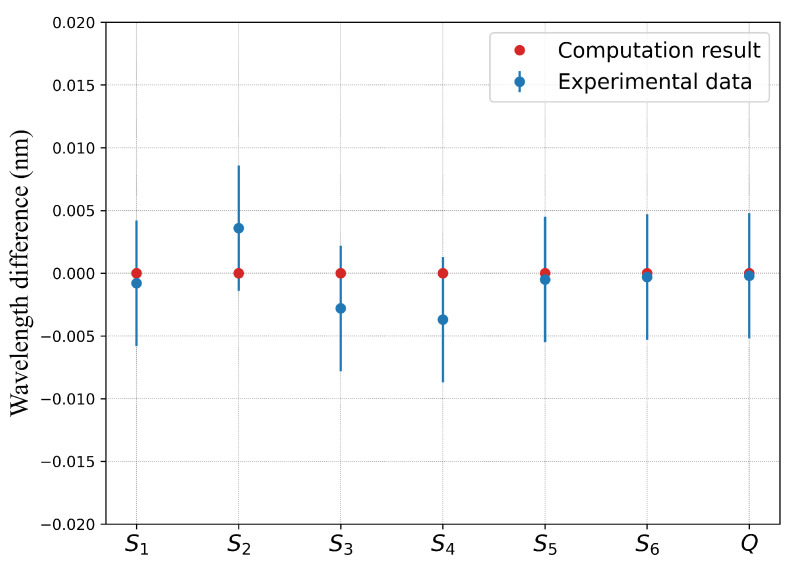
Bragg wavelength difference between the experimental data and the computations corresponding to the control points for FBG #1 with over-clad.

**Figure 6 sensors-22-01934-f006:**
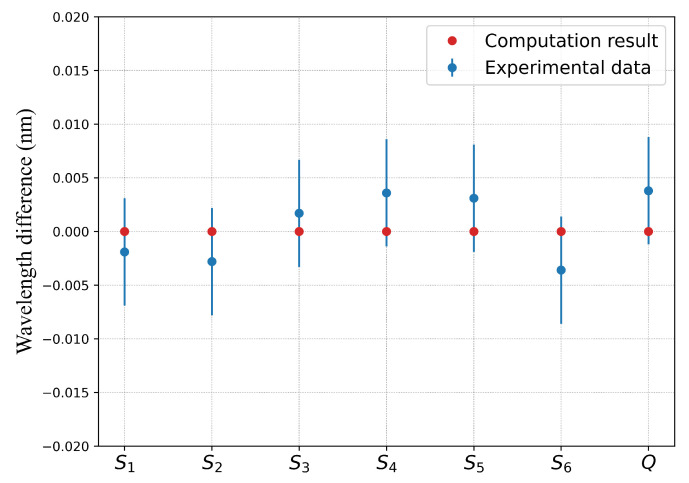
Bragg wavelength difference between the experimental data and the computations corresponding to the control points for FBG #2 without over-clad.

**Figure 7 sensors-22-01934-f007:**
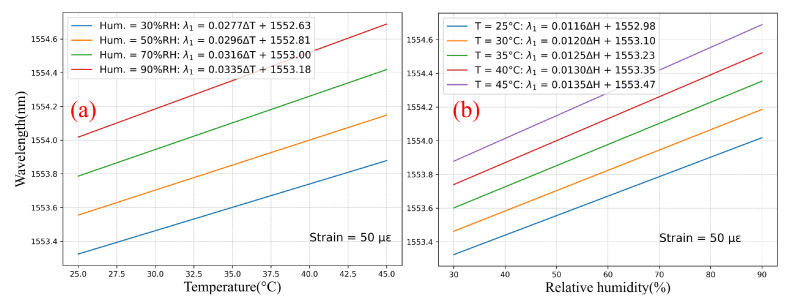
Computation of the CYTOP-FBGs with over-clad (FBG #1) response to (**a**) temperature, and (**b**) humidity within fixed strain conditions.

**Figure 8 sensors-22-01934-f008:**
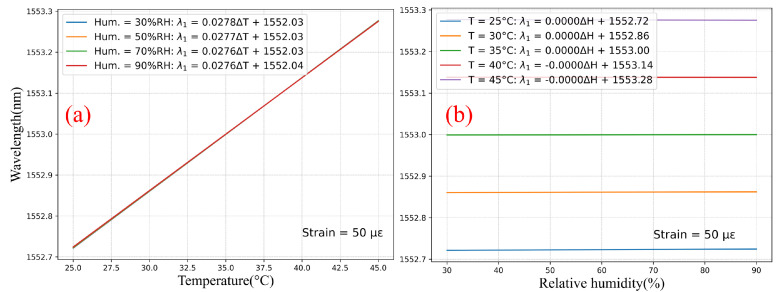
Computation of the CYTOP-FBGs without over-clad (FBG #2) response to (**a**) temperature and (**b**) humidity within fixed strain conditions.

**Table 1 sensors-22-01934-t001:** The levels of temperature, humidity and strain used in DoE.

Values	Symbol	Temperature ($X_{T}$)	Humidity ($X_{H}$)	Strain ($X_{S}$)
		°C	%RH	με
Low level	$X_{i,\min}$ ($x_{i}=-1$)	25	30	0.0
Mean level	$m_{i}$ ($x_{i}=0$)	35	60	77.5
High level	$X_{i,\max}$ ($x_{i}=+1$)	45	90	155.0
Half-range	$\Delta_{i}$	10	30	77.5

**Table 2 sensors-22-01934-t002:** Experimental matrix for a $2^{3}$ factorial design of FBG response to temperature, humidity, and strain in CYTOP fiber.

Trial	*I*	$x_{T}$	$x_{H}$	$x_{S}$	$x_{T}x_{H}$	$x_{T}x_{S}$	$x_{H}x_{S}$	$x_{T}x_{H}x_{S}$	λ
1	+1	−1	−1	−1	+1	+1	+1	−1	$\lambda_{a}$
2	+1	+1	−1	−1	−1	−1	+1	+1	$\lambda_{b}$
3	+1	−1	+1	−1	−1	+1	−1	+1	$\lambda_{d}$
4	+1	+1	+1	−1	+1	−1	−1	−1	$\lambda_{c}$
5	+1	−1	−1	+1	+1	−1	−1	+1	$\lambda_{e}$
6	+1	+1	−1	+1	−1	+1	−1	−1	$\lambda_{f}$
7	+1	−1	+1	+1	−1	−1	+1	−1	$\lambda_{h}$
8	+1	+1	+1	+1	+1	+1	+1	+1	$\lambda_{g}$

**Table 3 sensors-22-01934-t003:** The distribution of Bragg resonance wavelengths in 62.5 μm and 50 μm CYTOP fibers.

	62.5 μm CYTOP Fiber		50 μm CYTOP Fiber	
*m*	*n_m_*	*λ_m_* (nm)	*n_m_*	*λ_m_* (nm)
1	1.3405	1552.34	1.3406	1552.48
2	1.3395	1551.14	1.3392	1550.89
3	1.3384	1549.82	1.3379	1549.31
4	1.3373	1548.56	1.3365	1547.73
5	1.3362	1547.30	1.3351	1546.14
6	1.3351	1546.03	1.3338	1544.56
7	1.3340	1544.77	1.3324	1542.97
8	1.3330	1543.50	1.3310	1541.38
9	1.3218	1542.23	1.3296	1539.78
10	1.3207	1540.96		
11	1.3296	1539.69		

**Table 4 sensors-22-01934-t004:** Experimental results for the Bragg wavelengths at the eight corner points of Figure 1.

Wavelength (nm)	with Over-Clad (#1)	without Over-Clad (#2)
$\lambda_{a}$	1553.252	1552.641
$\lambda_{b}$	1553.806	1553.196
$\lambda_{d}$	1553.944	1552.643
$\lambda_{c}$	1554.618	1553.198
$\lambda_{e}$	1553.478	1552.891
$\lambda_{f}$	1554.032	1553.447
$\lambda_{h}$	1554.174	1552.896
$\lambda_{g}$	1554.838	1553.438

**Table 5 sensors-22-01934-t005:** Normalized main and cross sensitivities of FBG in CYTOP fiber with (Grating #1) and without over-clad (Grating #2).

Normalized Coefficient	Unit	Grating #1	Grating #2
$a_{0}$	nm	1554.018	1553.044
$a_{T}$	nm	0.306	0.276
$a_{H}$	nm	0.376	0.281 × 10^−13^
$a_{S}$	nm	0.113	0.124
$a_{TH}$	nm	0.029	−0.002
$a_{TS}$	nm	−0.130 × 10^−2^	−0.150 × 10^−2^
$a_{HS}$	nm	−0.250 × 10^−3^	−0.100 × 10^−2^
$a_{THS}$	nm	−0.130 × 10^−2^	−0.180 × 10^−2^

**Table 6 sensors-22-01934-t006:** Denormalized main and cross sensitivities of FBG in CYTOP fiber with (Grating #1) and without over-clad (Grating #2).

Denormalized Coefficient	Unit	Grating #1	Grating #2
$A_{0}$	nm	1554.018	1553.044
$A_{T}$	pm/°C	30.575	27.600
$A_{H}$	pm/%RH	12.525	0.950 × 10^−12^
$A_{S}$	pm/µε	1.456	1.603
$A_{TH}$	pm/(°C%RH)	0.096	−0.006
$A_{TS}$	pm/(°Cµε)	−0.002	−0.002
$A_{HS}$	pm/(%RHµε)	−0.110 × 10^−3^	−0.430 × 10^−2^
$A_{THS}$	pm/(°C%RHµε)	−0.500 × 10^−4^	−0.800 × 10^−4^

**Table 7 sensors-22-01934-t007:** Computational and experimental results of the Bragg wavelength of the control points in Figure 1 for gratings #1 and #2.

Trial	*X_T_* (°C)	*X_H_* (%RH)	*X_S_* με	*λ*_1,*Cal*._ (nm)	*λ*_1,*Exp*._ (nm)	*λ*_2,*Cal*._ (nm)	*λ*_2,*Exp*._ (nm)
$S_{1}$	35	30	77.5	1553.642	1553.641	1553.044	1553.042
$S_{2}$	45	60	77.5	1554.323	1554.327	1553.320	1553.318
$S_{3}$	35	90	77.5	1554.393	1554.390	1553.044	1553.045
$S_{4}$	25	60	77.5	1553.712	1553.708	1552.768	1552.770
$S_{5}$	35	60	0.0	1553.905	1553.905	1552.920	1552.922
$S_{6}$	35	60	155.0	1554.130	1554.130	1553.168	1553.172
Q	35	60	77.5	1554.018	1554.018	1553.044	1553.040

## Data Availability

Not applicable.

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
