# Peer review of "Properties of Fiber Bragg Grating in CYTOP Fiber Response to Temperature, Humidity, and Strain Using Factorial Design"

_sensors, 2022, doi:10.3390/s22051934_

Round 1

Reviewer 1 Report

The authors have proposed factorial method to investigate the humidity, temperature and strain in CYTOP fiber with and without clad. Although the investigation of these parameters in CYTOP fiber are not new, the method proposed is interesting.I suggest the following minor revisions for this manuscript.

  1. While the temperature and humidity range are proper the applied strain range (0 to 155µε) is too small. Any substantial reason for this ?
  2. What is the function of cover glass during inscription of FBG in fiber with over clad ?
  3. Which process was followed to remove the clad ?
  4. Getting multiple peaks in a multimode fiber is understandable. However, what is the reason for higher amplitude for λ1 . Why there is gradual increase in amplitude of FBG peaks from λ4 to  λ1.
  5. Why there are only 5 peaks in Fig.2 while there many Bragg resonance values in  Table 3.

Reviewer 2 Report

The work by Ying-Gang et al. investigated the properties of fiber Bragg grating in CYTOP fiber response to temperature, humidity, and strain using factorial design. The work is useful and important for the practical application of POF gratings. It is recommended to accept with some revisions as below:

  1. The motivation is not that clear. Why do you carry out this study? What is the key difference/feature for your contributions?
  2.  Please unify all the effective digits listed in the table.
  3.  The English is casual at some places. The authors should further check and revise it.
